# Controllability over stressor decreases responses in key threat-related brain areas

Chirag Limbachia[1], Kelly Morrow [1,2], Anastasiia Khibovska[1,3], Christian Meyer[4], Srikanth Padmala[5] & Luiz Pessoa [1,2,6,7 ✉]

Controllability over stressors has major impacts on brain and behavior. In humans, however, the effect of controllability on responses to stressors is poorly understood. Using functional magnetic resonance imaging (fMRI), we investigated how controllability altered responses to a shock-plus-sound stressor with a between-group yoked design, where participants in controllable and uncontrollable groups experienced matched stressor exposure. Employing Bayesian multilevel analysis at the level of regions of interest and voxels in the insula, and standard voxelwise analysis, we found that controllability decreased stressor-related responses across threat-related regions, notably in the bed nucleus of the stria terminalis and anterior insula. Posterior cingulate cortex, posterior insula, and possibly medial frontal gyrus showed increased responses during control over stressor. Our findings support the idea that the aversiveness of stressors is reduced when controllable, leading to decreased responses across key regions involved in anxiety-related processing, even at the level of the extended amygdala.

[1] Department of Psychology, University of Maryland, College Park, MD, USA. [2] Neuroscience and Cognitive Sciences program, University of Maryland, College Park, MD, USA. [3] Department of Psychology, Stony Brook University, Stony Brook, NY, USA. [4] Department of Human Development and Quantitative Methodology, University of Maryland, College Park, MD, USA. [5] Indian Institute of Science, Bangalore, Karnataka, India. [6] Maryland Neuroimaging Center, University of Maryland, College Park, MD, USA. [7] Department of Electrical and Computer Engineering, University of Maryland, College Park, MD, USA. ✉email: pessoa@umd.edu

Learned helplessness offers a dramatic illustration of the impact of controllability over biologically relevant outcomes[1]. Dogs learned to cross a barrier to avoid a stressor within a few trials. Yet, animals initially given inescapable stressors failed to later learn to escape it. Indeed, research over the past 50 years has revealed that uncontrollable electric shocks in rodents increase defensive reactions to threat including freezing, and subsequently impair instrumental learning in contexts where control is possible see[2,3]. In contrast, controllable stressor exposure is associated with the opposite profile: decreased freezing, increased social exploration, and improved instrumental learning.

At the neural level, the work of Maier and colleagues has uncovered important components of threat controllability, leading to a model in which the ventromedial prefrontal cortex (PFC) plays a central role is blunting the dorsal raphe nucleus' response to the stressor[4]. Critically, the serotonergic neurons of the dorsal raphe are only activated if the stressor is uncontrollable. Naturally, these two brain structures do not work in isolation but interface with several others, including the locus coeruleus, the bed nucleus of the stria terminalis (BST), the amygdala, and the periaqueductal gray (PAG). In particular, in the model by Maier and colleagues, the first two areas are conceptualized as inputs and the last two as outputs of the dorsal raphe[4].

Controllability research in rodents has inspired considerable work in humans[5–17], which also points to a prominent role for the ventromedial PFC during controllable conditions. However, a key gap in the human literature pertains to the effect of controllability on the responses to the stressors themselves, as opposed to anticipatory or learning-related dimensions. How do brain regions respond to a stressor event as a function of controllability? In humans, although some studies speak to this issue[5,6,11,14–17] (see also[18]), current knowledge is inconclusive and, critically, evidence for decreased responses during controllable conditions is very limited. Some studies have detected reduced responses during controllable trials only in a few voxels[14], only with specific statistical contrasts[17], or not at all[11]. In particular, the latter study used a within-subject yoked design and found comparable responses during controllable and uncontrollable painful stimulation, leading the authors to conclude that uncontrollable pain was not associated with more pain-related activation. In contrast, Salomons et al.[6] detected decreased responses during controllable conditions in the insula and amygdala, among other regions; Wood et al.[16] found decreases during controllable conditions in the mid/posterior hippocampus. Overall, it remains unclear how controllability affects stressor/pain-related responses in the brain.

Another important gap in the human literature refers to the limited knowledge of how controllability influences key threat-processing regions, notably, the BST[19], amygdala, and PAG[20]. Extensive debate exists about the extent to which regions such as the amygdala are modulated by high-level factors such as attention[21,22]. Determining how controllability impacts basolateral and central amygdala responses is thus important. Human studies have treated the amygdala as a unit and not characterized the participation of the functionally distinct basolateral and central amygdala subregions[23]. Understanding the roles of the central amygdala and BST is particularly important, as they form part of the so-called "extended amygdala"[19,24,25], a functional system that is central to fear- and anxiety-related processing.

To examine these questions, we built upon the moving-circles paradigm recently developed by our group[26], where two circles move around the screen, sometimes moving closer and at times moving away from each other. When the circles touch, participants are delivered a mild electric stressor (Fig. 1a; Supplementary Video 1). Note that circle movement, while smooth, has a high degree of unpredictability. For example, the circles might

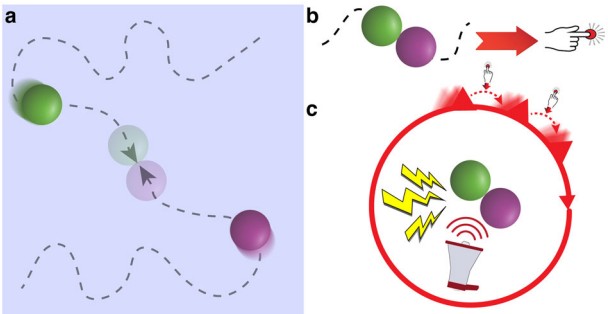

**Fig. 1 Moving-circles paradigm and controllability.** Paradigm schematic of a single trial with controllability where **a** shows a schematic representation of the display. When circles collided, a mild shock plug aversive sound was administered. For controllable participants, a button press moved the virtual wheel by 1/12 of the circle (**b** and **c**). The participant had to press the button multiple times to terminate the stressor. For uncontrollable participants, the virtual wheel moved the same number of times as that of the yoked participant. Uncontrollable participants were asked to press the button every time the wheel moved; the button press moved the virtual wheel a random amount (not shown). Stressor duration was matched between the two groups.

approach each other such that the stressor is more imminent, then retreat from each other for a period of time. In the present experiment, shock administration was accompanied by a loud buzzer-like sound, increasing the aversiveness of the overall stressor (shock plus sound). Two separate participant groups were scanned with functional magnetic resonance imaging (fMRI). In the controllable group, when the stressor was administered, its duration could be controllable by pressing a button to turn a virtual wheel (Fig. 1b, c). In the uncontrollable group, participants pressed a button but it did not have any relationship to stressor termination. In effect, the two groups were yoked such that the aversive stimulation experienced by participants was equated between them.

Whole-brain analysis with fMRI often lacks statistical power to uncover effects at the voxel level, which can lead to poor replicability[27]. Therefore, here we sought to focus on a set of regions of interest (ROIs) and leverage the strengths of Bayesian multilevel modeling[28,29] (BML) to estimate the effect of stressor controllability. One of the strengths of BML is that it allows the simultaneous estimation of multiple clustered parameters within a single model (for example, the effects at multiple schools within a district in an educational setting). In the present context, BML allowed the estimation of the effects across multiple ROIs simultaneously, or voxels simultaneously[30]. Among the advantages of this approach, information about the effect in one region/voxel can be shared across all regions/voxels (technically referred to as "partial pooling"). Another important feature is that correction for multiple comparisons is not needed as a single model is estimated[31].

To better understand how controllability of a stressor impacts responses to the stressor itself, the present study utilized a between-participants yoked fMRI task where participants were assigned to a controllable or uncontrollable stressor condition. Our analyses revealed that controllability decreased shock-plus-sound stressor-related responses in threat-related regions, including the BST and anterior insula, and increased responses in the posterior cingulate cortex and posterior insula. These findings provide further evidence that controllability reduces stressor aversiveness, raising the possibility that regions that exhibited increased responses in the control condition play important roles in this reduction.

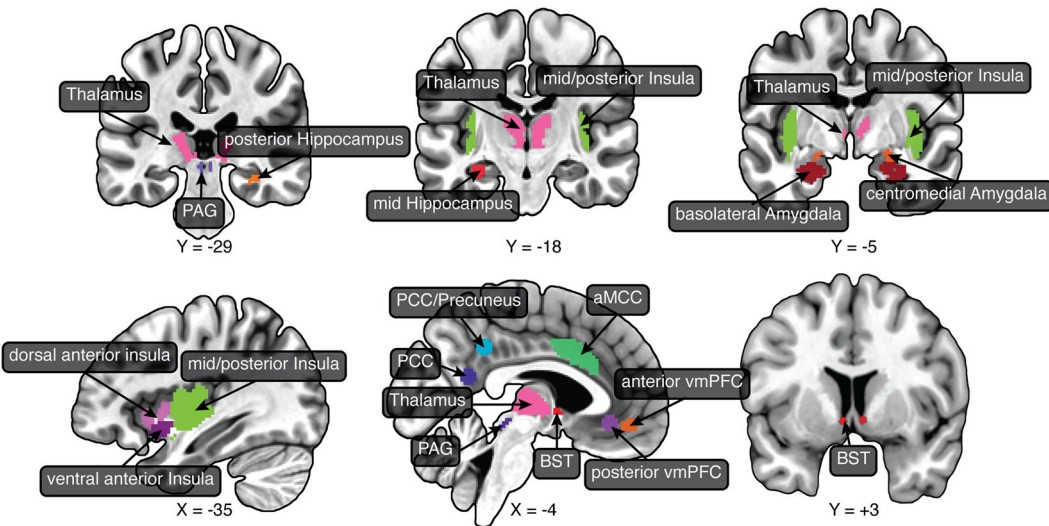

**Fig. 2 Regions of interest (ROIs).** The main analyses targeted a set of brain regions involved in threat processing. ROIs were defined anatomically, with the exception of the thalamus, vmPFC (both sites), PCC, and PCC/precuneus, which were defined based on separate functional datasets. aMCC anterior midcingulate cortex, BST bed nucleus of the stria terminalis, PAG periaqueductal gray, PCC posterior cingulate cortex, vmPFC ventromedial prefrontal cortex.

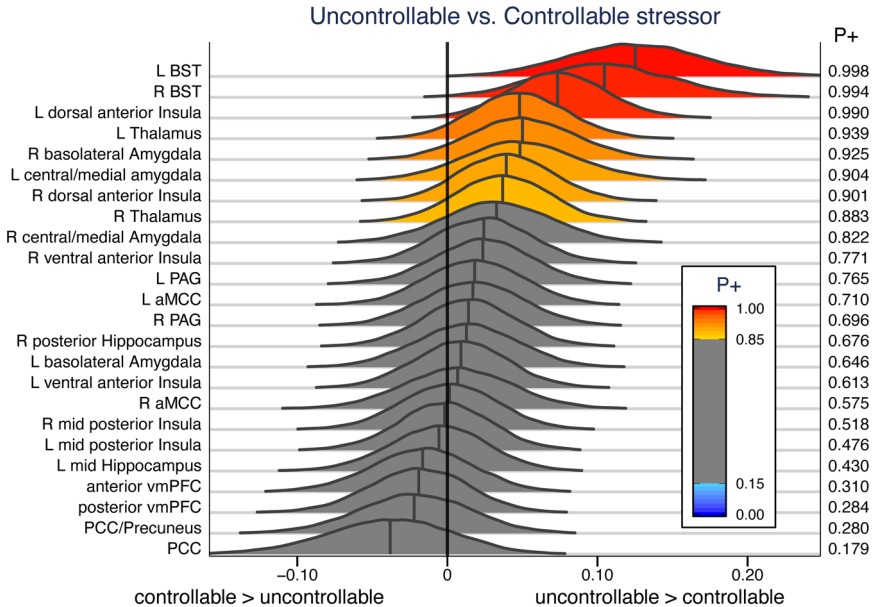

**Fig. 3 Posterior distributions of the effect of group membership.** Three regions of interest exhibited very strong evidence of stronger responses for uncontrolled participants: left/right BST and anterior dorsal insula. $P+$ indicates the probability that uncontrollable > controllable. aMCC anterior midcingulate cortex, BST bed nucleus of the stria terminalis, PCC posterior cingulate cortex, vmPFC ventromedial prefrontal cortex.

## Results

Most analyses utilized Bayesian methods, including simple tests of behavior, multilevel analysis for a set of 24 target ROIs (one representative time series per ROI obtained by averaging unsmoothed functional data to preserve spatial resolution), and multilevel analysis of voxelwise data from the insula. As Bayesian whole-brain voxelwise analysis was not computationally feasible at present, an additional standard voxelwise analysis was performed. For all analyses, the unit of interest was the yoked participant pair. In all Bayesian analyses, we report evidence for effects in terms of $P+$, the probability that the effect is greater than zero based on the posterior distribution: values closer to 1 provide evidence that the effect of interest is greater than zero (uncontrollable > controllable) while values closer to zero convey

support for a reverse effect (controllable > uncontrollable). We treat Bayesian probability values as providing a continuous amount of support for a given hypothesis; thus not dichotomously as in "significant" vs. "not significant".

**Behavior.** Stressor duration was equated between participant pairs (one participant from each group). Whereas participants from the controllable group executed button presses to terminate the stressor, button presses by uncontrollable-group participants had no bearing on cessation. Controllable participants produced considerably more button presses than uncontrollable participants ($173.16 \pm 18.01$ vs. $132.27 \pm 55.83$ [mean ± standard deviation] over the experimental session; $P+ = 1$). Thus, although

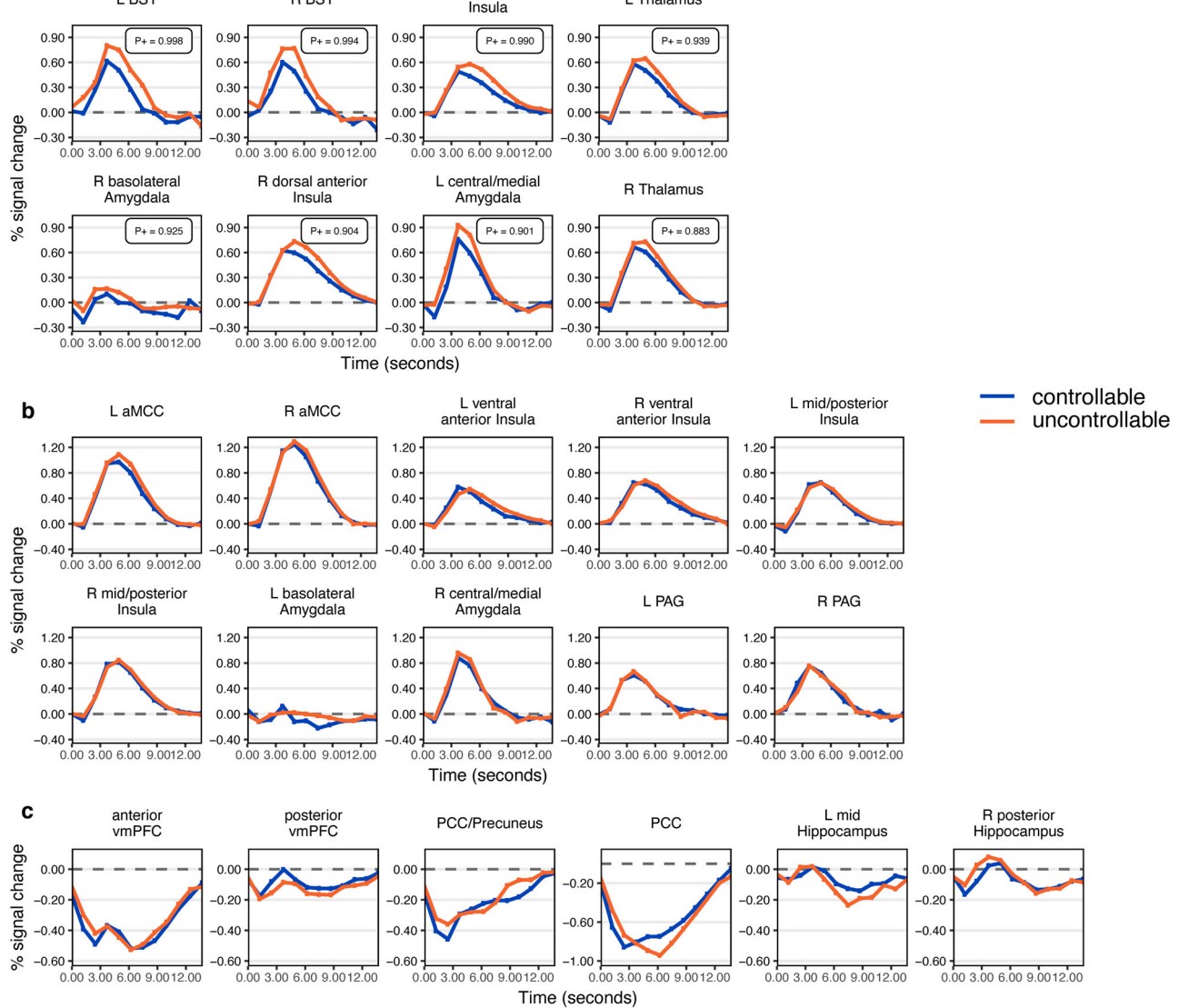

**Fig. 4 Stressor-related responses at regions of interest.** Unassumed shape analysis was used to estimate evoked responses to stressors for all ROIs. **a** ROIs with stronger evidence of a controllability effect. P+ values (P(effect > 0)) shown. **b**–**c** Responses for the remaining ROIs, with the responses to the stressor that were positive-going in **b** and those that were negative-going in **c**. Mid and posterior Hippocampus ROIs are also shown in **c**.

uncontrollable participants were instructed to press the button with every wheel turn, they did not.

Yoked pairs did not show evidence of difference in state (P+ = 0.49) or trait (P+ = 0.47) anxiety.

**Bayesian multilevel analysis at the level of region of interest.** We sought to determine the effect of controllability across a set of regions centrally involved in threat processing (Fig. 2). Key regions included the BST, basolateral and central/medial amygdala, and PAG. Here, we briefly motivate the additional ROIs investigated. Several theoretical frameworks consider the anterior insula to play a key role in anxiety[32,33], and the insula more generally is a key cognitive-emotional hub region[34,35,36]. The anterior/midcingulate cortex is important in the appraisal and expression of emotion[37,38]. Here, we focused on the anterior midcingulate cortex (aMCC), which has been suggested to be an area at the interface between emotion and motor interactions[39–41]. We also investigated the ventromedial PFC, which as

stated has been implicated in the modulation of responses to stressors[3]. We considered the anterior hippocampus, which has been discussed in the context of defensive behaviors[42–48], as well as mid/posterior hippocampus sites studied previously in the context of controllability[16]. The posterior cingulate cortex (PCC) and precuneus have been implicated in certain aspects of threat processing. For example, the PCC was engaged by "slowly attacking" threats[48], as well as more distal threat in our previous moving-circles study[26]. The PCC/precuneus was more engaged by a distal virtual tarantula[49] and distal threat in our moving-circles study[26]. Accordingly, we added sites along the posteromedial cortex to our set of ROIs so as to investigate the effect of controllability. Finally, we also considered the thalamus, which is increasingly understood to be a key region involved in threat processing[26,49,50].

Bayesian multilevel analysis was used to evaluate the effect of controllability in the 24 ROIs (Fig. 2). The linear model included covariates for the difference in the number of button presses, and both the average and difference in state/trait anxiety scores. The

mean state and trait score for a participant pair tries to capture the fact that some yoked pairs could have high/lower mean scores; the difference in state and trait scores captures the fact that individuals in a yoked pair could be relatively mismatched in terms of their anxiety scores (despite the procedure of matching them during recruitment), but as seen above there was no discernible evidence to support this.

The left and right BST, and the left dorsal anterior insula exhibited very strong evidence for increased responses for uncontrollable participants (P+ of ~0.99 or higher; Fig. 3). A few other ROIs exhibited a moderate amount of support for the same pattern of responses, particularly the left thalamus and the right basolateral amygdala, and the left central/medial amygdala. Noteworthy evidence for ROIs with controllable greater than uncontrollable responses was not observed at the ROI level (note that in such cases P+ should approach 0).

The analysis above was carried out by assuming the shape of the hemodynamic response to maximize statistical efficiency (see Methods). However, it is important to visualize stressor-related responses for at least two reasons. First, response visualization potentially aids in understanding the underlying processes. For example, are the responses transient and "positive going" (above baseline), or not as clearly transient and "negative going" (below baseline)? Second, visualization helps confirm that the assumed canonical response provides a reasonable approximation. Accordingly, responses to the stressor were estimated without assuming a canonical hemodynamic response (see Supplementary Methods). Many ROIs generated transient and robust responses to stressor events, such as the BST, PAG, and central/medial amygdala (Fig. 4). A relatively weak response was observed in the basolateral amygdala. Multiple ROIs along the midline generated negative-going responses, including PCC, precuneus/PCC, ventromedial PFC (two sites), in addition to the anterior hippocampus; the latter exhibited a more bimodal response. These results show that the canonical response provides a good model of responses for most ROIs, with a few exceptions including the PCC, precuneus/PCC, and anterior hippocampus (not because the responses were negative-going but because the shape appears to deviate from the canonical one; for example, for the PCC/precuneus the negative peak of the response occurred at 2.5 s, which is very early).

**Individual differences.** Evidence for individual differences in state/trait anxiety was limited (Supplementary Fig. 1). For example, for the left BST, the contribution of state anxiety difference was positive (P+ = 0.891).

**Brain–skin conductance correlation.** Skin conductance responses did not exhibit a difference between controllable and uncontrollable participants (P+ = 0.503). We investigated how the trial-by-trial relationship between skin conductance responses (SCRs) and brain responses varied as a function of controllability. We investigated this relationship for the left and right BST, which had the strongest effect of controllability (Fig. 5). We initially determined trial-by-trial responses to individual stressors, both in terms of skin conductance and fMRI signals. Trial-based responses were Spearman correlated for each individual, and compared between the two groups. The brain–SCR correlation was higher for uncontrollable compared to controllable participants in the right BST (P+ = 0.963); evidence was only modest for the left BST (P+ = 0.905).

**Bayesian multilevel analysis of insula voxels.** Presently, the Bayesian multilevel approach is not computationally feasible at the voxel, whole-brain level. Nevertheless, it can be extended to a

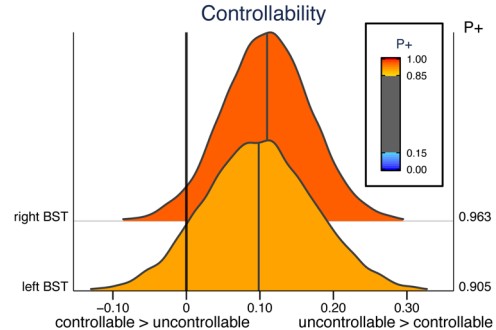

**Fig. 5 Trial-by-trial brain–skin conductance correlation and controllability.** Posterior distributions are shown for the left BST right BST. P+ indicates the probability that uncontrollable is great than controllable.

voxelwise approach within anatomical territories. Accordingly, we extended the approach to analyze voxels of the insula, a key region involved in threat/pain-related processing, among many other functions. In both hemispheres, voxels in both the dorsal and ventral anterior insula exhibited strong evidence for controllable < uncontrollable responses (Fig. 6). Notably, our analysis also uncovered clusters of voxels in the posterior insula where responses showed fairly strong evidence in the opposite direction: controllable > uncontrollable (P+ values <0.05).

**Standard whole-brain voxelwise analysis.** For completeness, we also ran an additional voxelwise analysis (standard, not Bayesian). Activation clusters were detected in the anterior insula, caudate, and putamen where responses decreased for controllable (Fig. 7a; Supplementary Table 1; see also Supplementary Fig. 2). The posterior cingulate cortex was the only cluster detected that exhibited a response that was greater for controllable participants (Fig. 7b).

As the ventromedial PFC has been implicated in controllability in the past, in an exploratory fashion, we relaxed our initial thresholding procedure and adopted a voxelwise cluster of 0.005 with a minimum cluster extent of 10 voxels. The contrast revealed additional controllable > uncontrollable clusters (Supplementary Table 2), including the medial frontal gyrus (Fig. 8).

## Discussion

In the present study, we investigated the effect of controllability on evoked responses to shock-plus-sound stressors in humans. Two circles moved around the screen and when they collided the aversive stimulus was administered. Participants in the controllable group could terminate the stressor by actively pressing a button; in contrast, in the uncontrollable group button pressing had no relationship to stressor termination. The experiment was designed by yoking the two groups such that participants experienced the same amount of aversive stimulation. Based on Bayesian multilevel analysis, we investigated responses in a targeted set of ROIs involved in threat processing centered on the BST, basolateral and central/medial amygdala, and PAG, in addition to other cortical and subcortical brain areas involved in threat processing. Additional Bayesian voxelwise analysis within the insula and standard voxelwise whole-brain analysis were performed, too. Next, we discuss some of our main findings.

Some studies found little[14] or no[11] evidence for reduced responses during controllable pain-related activation. In contrast, our analyses revealed reduced responses in the BST during controllable stressor administration. The BST is critically important for the processing of temporally extended and uncertain threat[19,51]. In humans, one study showed that responses

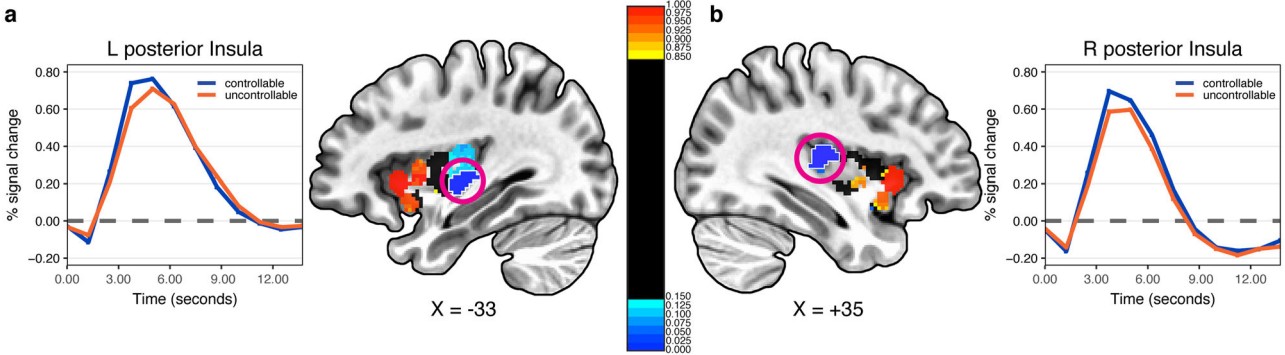

**Fig. 6 Bayesian multilevel voxelwise analysis of the left (a) and right (b) insula.** $P+$ indicates the probability that uncontrollable is greater than controllable. The estimated responses for left (**a**) and right (**b**) posterior insula regions (encircled blue regions) showed strong evidence for controllable > uncontrollable ($P+$ values <0.05).

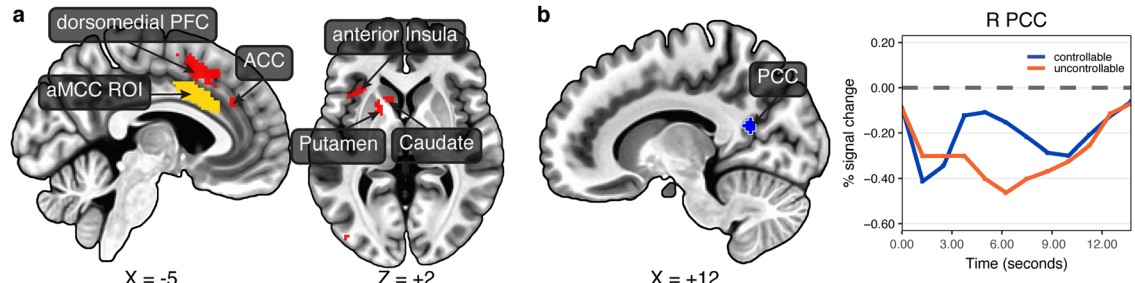

**Fig. 7 Voxelwise contrast of uncontrolled vs. controlled participants.** Maps were thresholded at 0.001 at the voxel level and 0.05 at the cluster level (13 voxels). **a** The anterior midcingulate cortex region of interest (yellow) is shown so that its location can be compared to the voxelwise activation clusters that exhibited greater responses in uncontrollable participants. **b** The posterior cingulate was the only cluster that showed a greater response in controllable participants. ACC anterior cingulate cortex, aMCC anterior midcingulate cortex, PCC posterior cingulate cortex, PFC prefrontal cortex.

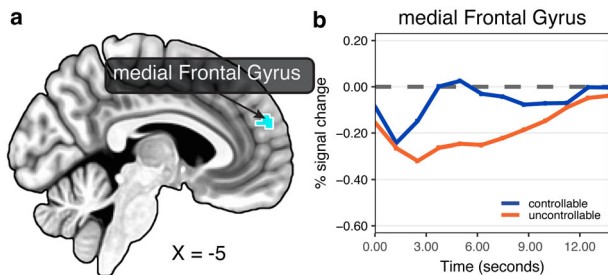

**Fig. 8 Exploratory voxelwise analysis. a** Voxelwise threshold at 0.005 and cluster extent of at least 10 voxels: controlled > uncontrolled. **b** Estimated response of the medial frontal gyrus cluster based on unassumed shape analysis (% signal increase).

increased parametrically with threat (a virtual tarantula) proximity[49]. Although data processing employed considerable spatial smoothing, the parametric effect appeared to include the BST. In our previous study using the moving-circles paradigm (where stressor delivery was uncontrollable), we also detected a parametric effect of threat proximity on BST responses. This latter study is part of a growing literature in humans that has steadily improved imaging parameters and procedures to image this technically challenging region; for example, refs. [52–54].

In the present study, the BST exhibited strong transient response to the stressor itself. Critically, responses differed across participant group, being reduced for those having control over the stressor. How can these findings be interpreted? The results

from the tarantula study[49], as well as the threat-proximity effect in our previous moving-circles study[26], indicate that more imminent stressors engage the BST more strongly. In addition, in the tarantula study, participants' "experienced fear" ratings increased substantially when the tarantula was perceived as near. Thus, it is reasonable to assume that stressors perceived to be more powerful are associated with enhanced BST responses. Accordingly, in the current study, the decreased responses in the BST could reflect the diminished aversiveness or perceived impact of the stressor in controllable participants. Our findings add to the body of data on the extended amygdala comprised of the BST and central amygdala[25,55], in particular the question of the functional roles of these two areas. Whereas evidence for a controllability effect in the left/right BST was very strong, it was moderate for the left central/medial amygdala and very weak for the right central/medial amygdala. Our analyses also uncovered a relationship between trial-by-trial brain responses and SCRs. Specifically, the correlation between trial-by-trial responses was higher for uncontrollable participants in the right BST. These results provide support for a close relationship between stressor-related activation in the BST and autonomic responses indexed via skin conductance.

The anterior insula (both dorsal and ventral) produced a pattern of responses that paralleled those in the BST: decreased responses for controllable participants. Although the anterior insula is very functionally diverse[56], it has a prominent role in threat processing, especially when uncertainty is involved[32,33]. The anterior insula is often conceptualized in terms of dorsal and ventral sectors, with the former more closely tied to cognition and the latter with emotion[57], although these sectors are quite

functionally diverse[36]. As in the case of the BST, decreased responses might reflect the diminished aversiveness or perceived impact of the stressor in controllable participants. These findings are important because the anterior insula has been suggested to be a cognitive–emotional hub region central to emotional awareness and involved in answering the following question: how do you feel now?[34].

Unlike previous animal and human studies[3], we did not detect the participation of the ventromedial PFC but uncovered three sites that may play a comparable role as attributed to this region when stressor controllability is possible, the medial frontal gyrus, the posterior cingulate cortex, and the posterior insula, all of which produced greater responses in controllable participants. The medial frontal gyrus cluster is particularly relevant given previous human studies that have observed anterior medial PFC sites during adaptive coping, for example[8,11,58]. As the medial frontal gyrus cluster was observed during exploratory analyses, we consider the evidence more tentatively (nevertheless, the cluster contained 30 voxels and, at a 0.005 voxelwise threshold, clusters of 27 or more voxels are significant at the cluster level of 0.05). In studies of threat processing, the PCC/precuneus is more strongly engaged during less anxious states[26,49,59]. In both the virtual tarantula and the moving-circles paradigm[26,49], this sector exhibited parametric threat effects, such that responses increased for less proximal threat; conversely, responses decreased for more proximal threat. In the voxelwise analysis, an activation site in the right PCC showed an effect, such that responses increased for controllable participants. The BML voxelwise analysis uncovered voxels in the posterior insula where controllable responses were greater than uncontrollable. These results are noteworthy given evidence for controllability-related "safety signals" in this area in rats[60]. Intriguingly, the safety signal from the posterior insula attenuated markers of neural activity (Fos expression) in the BST and basolateral amygdala. These findings are reminiscent of the reduced responses of the BST during controllable conditions (and to some extent in the right basolateral amygdala).

As stated in the Introduction, a model of controllability centered on the dorsal raphe has been proposed by Meier and colleagues[4]. In their proposal, the ventromedial PFC provides "behavioral control", thereby blunting the impact of aversive stimulation by inhibiting the dorsal raphe when control is possible. They propose that the locus coeruleus and the BST provide inputs to the dorsal raphe, while the central amygdala and the PAG are its output targets. Our results cannot be directly compared to their model, especially because our paradigm differed in important ways compared to the work by Meier and colleagues[4] (e.g., threat approach/retreat). Nevertheless, it is informative that we did not detect an effect of controllability in the PAG and only a modest one in the central/medial amygdala, which their model would predict when a stressor is controllable. Instead, the strongest effect of controllability was observed in the BST (together with the anterior insula), which we tentatively interpret as associated with the decreased aversiveness of the stressor during controllable conditions.

We suggest that one of the advantages of the Bayesian analysis performed here is that the results can be interpreted not in terms of a binary decision (significant vs. not significant) but in a graded fashion. Accordingly, the evidence in support of a controllability effect was very robust in the BST, but moderate for the left central/central amygdala and left thalamus, for example. These regions are, of course, extremely important for threat processing, and we believe discussing them in the context of the strength of evidence is valid scientifically, and preferable to dichotomizing results based on a fixed threshold such as 0.05 in standard null hypothesis testing. The Bayesian multilevel approach we applied at the ROI and voxel levels illustrates how information sharing across ROIs/voxels can be used as an alternative to traditional univariate approaches, which lead to severe multiple-testing and correction issues under standard inferential testing. In the Bayesian multilevel approach, a single mathematical model can be defined for the entire data, if computational resources are available. In addition, as the estimate at each ROI/voxel is obtained in the context of every other ROI/voxel, estimates are more conservative (technically, they tend to be pulled toward the overall mean[30] which can also aid in replicability).

Controllable participants pressed the button to stop the virtual wheel more often than uncontrollable participants. Is it possible that some of our results were driven by differences in button presses? Bearing this concern in mind, to account for that possibility, the data were analyzed with a model that included a parametric regressor for the number of button presses at the participant level. As an additional protection, we further added a covariate for the difference of button presses between yoked participants (at the group level). Therefore, whereas we cannot exclude the possibility that motor responses contributed to group differences, we believe it is highly unlikely that it was a dominant force given our two-pronged approach. Furthermore, at the yoked-pair level, BML captured some contribution of button presses to our contrast in the BST, but in the opposite direction that would have contaminated results. Specifically, the button-press covariate was larger for controlled participants, the group that pressed less often (Supplementary Fig. 1).

In summary, in the present paper we investigated how controllability alters the processing of a shock-plus-sound stressor in the human brain. Using a between-group yoked design, participants in the controllable condition were able to press a button to turn a virtual wheel to stop the stressor. Although stressor delivery was identical for the two groups, controllability robustly decreased responses in the bed nucleus of the stria terminalis and anterior insula. Our Bayesian multilevel modeling approach also revealed moderate evidence for response dampening in the basolateral amygdala, central/medial amygdala, and thalamus. When combined with other studies of threat processing, our findings support the idea that aversiveness of the stressor is reduced when it is controllable leading to decreased responses in key anxiety-related brain regions. Finally, we found three regions that exhibited greater responses in controllable participants: the medial frontal gyrus, the posterior cingulate cortex, and the posterior insula. We suggest that these sites are potentially important in reducing the impact of aversive stressors during controllable conditions.

## Methods

**Participants**. One hundred and twenty-six participants (63 females, ages 18–30 years; average: 20.87, STD: 2.56) with normal or corrected-to-normal vision and no reported neurological or psychiatric disease were recruited from the University of Maryland community. The project was approved by the University of Maryland College Park Institutional Review Board and all participants provided written informed consent before participation. Data from four participants were not used (one due to claustrophobia; one due to multiple episodes of falling asleep; and two given that the experiment was terminated before matching participants could be recruited (see below).

**Procedure and stimuli**. Two circles moved around on the screen in a smooth fashion. When they collided with each other, an unpleasant mild electric stressor was delivered together with an aversive sound. The algorithm to generate the motion of the circles generated approach and retreat periods of varying durations (2–9 s), but with most of them lasting more than 6 s of approach or retreat. The total amount of time during which the circles were approaching or retreating was equated. Overall, circle motion was designed to be smooth but still relatively unpredictable in terms of the pattern of approach/retreat. In particular, multiple instances of "near misses" were incorporated such that the circles retreated once they were close to each other (see below). The paradigm was developed with multiple study goals in mind, including related to threat proximity[26] and threat

dynamics[61]. Here our focus was on the stressor delivery component and the role of controllability.

Visual stimuli were presented using PsychoPy (http://www.psychopy.org/) and viewed on a projection screen via a mirror mounted to the scanner's head coil. Each participant viewed the same sequence of circle movements. The experiment included 6 runs (8 min each), each of which had two 3-min blocks (2 participants had only 5 runs; 3 participants had only 4 runs; the matching runs were eliminated from the yoked participant's data). During each block, the circles appeared on the screen and moved around for 180 s; blocks were separated by a 30-s off period during which the screen remained blank. A 15-s blank screen was presented at the beginning of each run; a blank screen was presented at the end of the run.

In each block, circles collided 0–3 times. A total number of 25 collisions occurred during the course of the experiment (for participants with only 4 and 5 runs, 17 and 20 collisions, respectively). To minimize any learning of motion patterns, participants experienced on average 4 stressors and 7 near-misses per run, so they could not determine when the circle approach would culminate in stressor delivery (a near-miss was defined as the circles approaching each other and then retreating when the edges of the two circles were 1.5 circle or less diameters away). In addition, because all participants experienced the same motion paths, the two groups were equated in this respect.

Each collision resulted in the delivery of an electric stressor and of an aversive sound stimulus. The electric stressor (comprised of a series of current pulses at 50 Hz) was delivered by an electric stimulator (Model number E13-22 from Coulbourn Instruments, PA, USA) to the fourth and fifth fingers of the non-dominant left hand via MRI-compatible electrodes. Stressor delivery (shock + sound) was applied for 1.37 s (median), ranging from 0.16 to 4.27 s (although the bulk of them ranged from 0.5 to 3 s). Electrical stimulation was administered in 500 ms on/500 ms off alternations. Thus, a shock for 1.75 s was on for 500 ms, off for 500 ms, and then on for 250 ms. In addition, when on, the waveform of the stimulator administered pulses of 10 ms of electrical stimulation alternating with 10 ms of no stimulation.

To calibrate the intensity of the stressor, each participant was asked to choose his/her own stimulation level immediately prior to functional imaging, such that the stimulus would be "highly unpleasant but not painful." After each run, participants were asked about the unpleasantness of the stimulus in order to re-calibrate stressor strength, if needed. Each collision also resulted in the playing of a buzzer-like aversive sound that was on for the duration of the stressor stimulus. The loudness of the sound was set to be "loud but not uncomfortable" by each participant (but never exceeded 85 dB).

Skin conductance response (SCR) data were collected using the MP-150 system (BIOPAC Systems, Inc., CA, USA) at a sampling rate of 250 Hz by using MRI-compatible electrodes attached to the index and middle fingers of the non-dominant left hand. Due to technical problems SCR data were not available for 4 participants (the corresponding data for the yoked participants were also not utilized).

**Virtual wheel procedure and participant yoking**. When a stressor was delivered, the screen turned white and a red dial appeared around the circles (Fig. 1c). Subjects in the controllable group were told that moving the wheel allowed them to terminate the stressor; specifically, they were told the duration of shock could be reduced if they pressed the button frequently and fast enough. The wheel turned clockwise by 30 degrees (1/12 of the circle) with every button press. The number of rotations required to terminate the stressor started at one and increased after each successful stressor termination up to 12 total button presses. This procedure was implemented so as to parallel the rodent procedure adopted by Maier and colleagues[4] (see ref. [62]). The procedure had the effect of increasing the stressor duration as the experiment progressed. If a subject failed to escape a stressor in a run, data from that run was not utilized (one participant had 1 run discarded).

Subjects in the uncontrollable group experienced the exact same duration of stressor delivery as yoked participants in the controllable group. Subjects in the uncontrollable group saw the dial rotate on its own during stressor delivery. The timing of the rotation of the dial was matched to that of the yoked controllable participant, but the amount of rotation was random to avoid any sense of progression. Participants were told to press the button whenever they saw the dial rotate, so as to maintain the number of button-presses closely matched to that of the yoked controllable participants; specifically, they were asked to press the button to test their motor responding.

**Regions of interest**. We focused on the following ROIs, which are shown in Fig. 2. Subcortically, we targeted the following regions: Amygdala (2 sectors), BST, PAG, and thalamus. Cortically, we targeted the following regions: Insula (3 sectors), anterior midcingulate cortex, PCC, PCC/precuneus, and ventromedial PFC (2 sites).

The amygdala ROIs were defined based on the anatomical masks by Nacewicz[63], which we used in an earlier version of the moving-circles paradigm[64]. We created amygdala ROIs for the central/medial amygdala (the central amygdala is too small to be separated from the medial amygdala with the present resolution) and for the basolateral amygdala (union of BLBM and La masks).

The PAG ROI was defined anatomically based on the definition by Erza et al.[65]. However, as multiple voxels from the original mask are adjacent to the

cerebrospinal fluid aqueduct, we eroded the mask to exclude these voxels, as follows. Using a separate fMRI study[26], we evaluated the signal quality of aqueduct and non-aqueduct voxels. The aqueduct contained no voxels with a signal-to-noise ratio (SNR) greater than 20, whereas SNR within the original Ezra mask spanned from 16 to ~40. To exclude aqueduct-containing voxels, we eliminated those with SNR less than 25, effectively staying clear of the aqueduct.

The BST ROI was defined anatomically according to the probabilistic mask of the BST (at 25% threshold) reported by Blackford and colleagues[66], as used in our previous study[26].

The anterior hippocampus was defined based on a recent parcellation of subcortex[67]. The study by Wood and colleagues[16] revealed controllability effects in the mid/posterior hippocampus. Based on their study, we defined a left mid hippocampus ROI by creating the intersection of a 5-mm radius sphere centered on the coordinate they provided and the posterior hippocampus ROI of Tian and colleagues[67]; any intersection with the anterior hippocampus was eliminated. A right hippocampus ROI was created via the intersection of a 5-mm radius sphere centered on the coordinate they provided and the posterior hippocampus ROI of Tian and colleagues[67].

The insula ROIs were defined based on the masks by Faillenot et al.[68]: dorsal anterior insula (union of the anterior short gyrus and the middle short gyrus), ventral anterior insula (anterior inferior cortex), and middle plus posterior insula (union of the posterior long gyrus, anterior long gyrus, and posterior short gyrus).

The anterior midcingulate cortex was defined based on the mask from Destrieux et al.[69]. Additional cingulate/midline ROIs and medial PFC ROIs were defined based on functional data from separate studies. In all cases they were spherical ROIs with 5 mm radius. A ventromedial PFC ROI (called posterior ventromedial PFC) was defined based on the coordinates provided by Boeke et al.[13]. A second ventromedial PFC ROI (called anterior ventromedial PFC), as well as PCC and PCC/precuneus ROIs, were based on functional data from our previous moving-circles study[26], specifically, by evaluating the effect of threat proximity (contrast of farther vs. closer circles); in these three cases, activation was stronger when the circles were farther (the opposite of a region such as the anterior insula).

**Statistics and reproducibility**. Statistical analysis was performed with a two-level procedure: estimation of regression coefficients at the individual level, followed by group analysis. Participant-level estimation adopted a standard multiple regression approach. The main group-level analysis followed a Bayesian multilevel modeling framework[31], which has been developed recently for fMRI data[30]. This analysis was performed at the level of ROIs. Bayesian multilevel group analysis is not computationally feasible for whole-brain voxelwise analysis but we applied it to voxels within the anatomical insula (see below). For completeness, we also performed a parallel standard voxelwise group-level analysis that compared uncontrollable and controllable groups via a paired $t$ test (given participant yoking).

*Participant level*. Preprocessed fMRI data of each participant was analyzed using multiple linear regression implemented in AFNI's 3D Deconvolve program. Two separate processing streams were employed. For ROI-level analysis, unsmoothed data were employed; for voxel-level analysis, functional data were blurred using a 4 mm full-width half-maximum (FWHM) Gaussian filter (spatial smoothing was restricted to gray-matter mask voxels). The intensity of each voxel was normalized to a mean of 100 (separately for each acquisition run) in both processing streams.

In our previous studies using similar versions of the moving-circles paradigm, we observed effects of direction (approach vs. retreat) that varied as a function of circle proximity (defined based on the Euclidean distance between the circles); in other words, an interaction. Accordingly, in the present analysis, we subdivided the entire range of distances between circles into proximity segments: "near" captured periods during which the circles were in relatively close range (>33% of the maximum possible distance); "far" captured the remaining time. Note that, by definition, a stressor, which was the event of interest in our analyses, only occurred when proximity was "near". Every stressor event varied in duration to some extent (range: 160 to 4270 ms). Accordingly, a regressor was defined based on stressor onset times and their durations.

To account for other contributions to fMRI signals, three additional regressors were considered: direction (+1 during approach, −1 during retreat), speed (temporal difference of proximity values), and the interaction between direction and speed (mean centered). Direction and speed regressors were set to zero during stressor administration. These three regressors were defined for the "near" period and, separately, for the "far" period. Given our interest in characterizing stressor-related responses, here we were only interested in the "near" period. Note that a stressor event always followed an approach period, but the relationship between approach segments and the stressor was variable (see also collinearity below). In particular, when the circles approached each other, even in the near space, they frequently altered course and retreated from each other. Of these approach-to-retreat changes, multiple of them were designed to be "near misses" so that the circles came very close together before retreating.

When the circles touched, the stressor was administered, and participants in each group pressed the button (typically multiple times). Although participants were yoked to match stressor duration, the number of button presses could potentially vary between yoked pairs. Accordingly, in addition to the stressor

regressor described above, a second regressor was employed that modeled a participant's number of button presses. This parametric regressor had the same onset and duration as the basic stressor regressor, but was mean centered around the mean number of button-presses to reduce collinearity (mean centering was done on a run-by-run basis because the required number of button-presses increased along the session). Overall, the parametric regressor captured signal variance that was linked to the number of button presses, thereby minimizing potential differential contributions of the participant's overt behavior.

All the above regressors were convolved with a standard hemodynamic response as defined by the gamma variate model[70]. In addition, six motion parameters (three translational and three rotational) and their discrete temporal derivatives were included in the model to account for head motion. To account for baseline and drift in the MR signal, linear and non-linear polynomial terms (up to fourth order) were also included in the model. We evaluated collinearity and computed the variance inflation factor associated with all variables[71]. The maximum value was 1.9, revealing that regressor correlation did not unduly compromise parameter estimation. In particular, the parametric variable based on the number of button presses did not lead to high regressor correlation given that it was mean centered (the correlation of this regressor with others was <0.5). Note that although suggested variance inflation factor cutoffs are necessarily crude, a value of 10 is often described as reason for concern; more conservative suggestions consider a value of 2.5 or higher of possible concern[72]. However, as discussed by Mumford et al.[73], collinearity is not a major concern in two-level statistical analyses such as with fMRI data.

*Bayesian multilevel statistical analysis: region of interest.* The null hypothesis significance testing (NHST) framework has come under increased scrutiny in recent years. In particular, the hard threshold of 0.05 has come under attack, with reasonable researchers calling for stricter thresholds[74] or, conversely, for the dichotomous use of *p*-values to be abandoned[75]. Our own approach, which is described in more detail elsewhere[76], does not consider a binary threshold ("significant" vs. "not significant") to be a productive way to evaluate statistical evidence. Accordingly, in the present paper, wherever possible, we employed Bayesian statistical analysis (all analyses except the voxelwise case).

The Bayesian framework attempts to estimate the probability of a research hypothesis $H$ given the data, $P(H|\text{data})$. The framework is not typically formulated to generate a binary decision (e.g., "real effect" vs. "noise") but instead to obtain the entire probability density distribution associated with $P(\theta|\text{data})$, where $\theta$ is the parameter being estimated. Such posterior distribution allows the quantification of, for example, $P(\theta > 0 \mid \text{data})$, the area under the curve above 0 which we call $P+$. Values closer to 1 provide evidence that the effect of interest (e.g., mean, difference of means, etc.) is greater than zero (conditional on the data, the prior distribution, and the model); values of $P+$ closer to zero convey support for a negative effect (for example, $P+ = 0.01$ indicates that the probability of the effect being positive is only 0.01, which implies that the probability of it being negative is 0.99).

The posterior distribution provides a summary description of the likelihood of observing parameter values given the data, so it naturally conveys variability. Some authors use cut-off points to summarize "strong" (e.g., >0.975), "moderate" (e.g., >0.95), or "weak" (e.g., >0.9) evidence, but we encourage a more qualitative approach without making decisions in terms of "passes threshold" versus "fails to pass threshold". For example, although a $P+$ value of ~0.9 is relatively "weak", if the region in question is of theoretical importance, discussing the finding would be relevant. Loosely speaking, whereas one would not necessarily discuss every region with moderate or weak evidence, discussing those that are conceptually relevant would be appropriate.

Note that we do not employ Bayes factors, which some have advocated as a potential feature of Bayesian modeling. Because Bayes factors consider the probability of "null" effects (e.g., a mean of zero) versus an "alternative" effect (e.g., a mean different from zero), we believe its use is often problematic, because formulating the problem in terms of "null" effects of exactly zero is often unrealistic (because effects of experimental manipulations are seldom zero), thus largely inflating the evidence for the alternative hypothesis (creating "large" Bayes factors). See Chen et al.[76] for further discussion. Finally, given that we do not view thresholding as adequate, the $P+$ probability values that we provide are (by definition) "one-sided". For readers who absolutely insist on comparing $P+$ values with standard cutoffs that are "two-sided", they should bear in mind our definition.

We employed a Bayesian multilevel analysis at the level of ROIs[77]. ROI-level data employed the average-across-voxels time series for each ROI; thus, 24 representative time series were employed. Averaging was performed on non-spatially smoothed data to avoid degradation of spatial resolution. In the approach adopted, the data from all ROIs are included in a single multilevel model that evaluates the effects of interest. By doing so, the contributions to fMRI signals of subject-level effects (i.e., subject effect across conditions), and ROI-level effects (i.e., ROI effect across subjects), can be accounted for in a model that simultaneously ascertains the effect of controllability. The "output" of the Bayesian multilevel model comprises only one overall posterior that is a joint distribution in a high-dimensional parameter space; thus, no correction for multiple comparisons is needed[31]. For summary purposes, posteriors of the effects for every ROI can be plotted separately; but they are not independent and technically are simply marginal distributions (that is, projections along particular variables). For formal

details of the approach adopted here, please refer to Chen et al.[30]; for a less technical exposition, see Chen et al.[76]. In contrast, based on the standard testing approach, the effect at each region is estimated independently from that at other regions, which calls for a correction for multiple comparisons.

The central question of interest was to compare the difference in stressor responses between the two groups:

$$\Delta_p = \text{stressor}_p^{\text{UNCTL}} - \text{stressor}_p^{\text{CTL}}$$

where the superscript UNCTL and CTL refer to the two groups, $p$ indexes paired/yoked participants, and stressor refers to the parameter estimates from multiple regression from the first-level analysis. However, the difference between the two groups could be itself influenced by factors that differentially affected them, including differences in state and trait anxiety between yoked participants. In addition, differences in button-presses could also influence group differences. Thus, we considered these three types of difference as covariates to be included in the model. In addition to differences in state/trait scores, their sum (or average) could also be a factor. To see this, consider that a yoked pair with higher average state anxiety could potentially produce larger differential stressor-related responses than a pair with lower average state anxiety. Accordingly, we included average state and trait anxiety scores as covariates in the model to evaluate how the effect of controllability was affected by them. Thus far, the model can be summarized in Eq. 1:

$$\Delta_p = b_0 + b_1 \text{StateAnxiety}_p^{\text{DIFF}} + b_2 \text{StateAnxiety}_p^{\text{AVG}}$$
$$+ b_3 \text{TraitAnxiety}_p^{\text{DIFF}} + b_4 \text{TraitAnxiety}_p^{\text{AVG}} + b_5 \text{ButtonPress}_p^{\text{DIFF}} + \epsilon \quad (1)$$

where $p$ indicates a pair of yoked participants, DIFF indicates the difference score, and AVG indicates the mean score. In our sample, the correlation between state and trait anxiety was 0.5; between state average and state difference was 0.05; and between trait average and trait difference was −0.22.

Within a multilevel framework, we extended the model above to simultaneously consider the contributions of both participant and ROI effects. This allowed us to model a term (intercept) per participant pair (capturing the yoked-pair's specific contribution relative to the overall intercept), and a term (intercept) per ROI (capturing the ROI-specific contribution relative to the overall intercept), in addition to an overall intercept term. Likewise, each covariate could be modeled via an overall slope plus a slope that was ROI specific. In the terminology of linear mixed effects models, such "varying effects" models, include both "varying intercepts" and "varying slopes"[29,31]. Although linear mixed effects models can be very powerful, parameter estimation can be problematic and/or not possible. However, a Bayesian formulation seamlessly allows the efficient estimation of parameters, while providing interpretability in terms of evidence. The appendix describes the model formally for completeness, including the weakly informative priors employed, which had negligible impact on the posterior distributions.

All BML models were implemented using rstan (https://mc-stan.org/users/interfaces/rstan) which is the R interface to the probabilistic Stan language (https://mc-stan.org/). Stan estimates posterior distributions with state-of-the-art Monte Carlo Markov Chain methods. Estimation convergence was evaluated with $\hat{R} < 1.1$ (all values near 1.0).

*Bayesian multilevel statistical analysis: voxelwise within the insula.* Whole-brain voxelwise analysis via BML modeling was computationally prohibitive with current resources. Here, we analyzed voxels of the left and right insula, separately, according to the same multilevel strategy outlined above. For each of the two hemispheres, the Bayesian model estimated a single multi-dimensional posterior distribution[31].

Because the insula was anticipated to be functionally heterogenous, we first subdivided the insula of each hemisphere into anatomically defined three ROIs: dorsal anterior, ventral anterior, and mid-plus-posterior insula (see Supplementary Methods). Next, each ROI was subdivided into sub-ROIs containing comparable number of voxels. To do so, we simply clustered voxels based on the $xyz$ MNI coordinates (via k-means clustering). The 941 voxels of the left insula were subdivided into 11 sub-ROIs (each containing 67–113 voxels); the 986 voxels of the right insula were subdivided into 10 sub-ROIs (each containing 74–145 voxels). The multilevel model simultaneously estimated the contributions of the subject, sub-ROI, and voxel.

*Additional Bayesian tests.* To evaluate group differences of button presses, we employed the R brms package[78] in which builds upon Stan (https://mc-stan.org/). For the button-press analysis, a simple intercept model was evaluated:

$\Delta_p = \text{buttonpress}_p^{\text{UNCTL}} - \text{buttonpress}_p^{\text{CTL}}$ where button press refers to the number of button presses (other notation as in Eq. 1). We assumed a prior distributed as a $t$ distribution which accommodates data skew and outliers well[79]. Analogous models were used to test for differences in state and trait anxiety.

We also evaluated the relationship between trial-by-trial responses in select ROIs (left and right BST, and left dorsal anterior insula) and trial-by-trial responses in skin conductance. Brain–SCR Spearmann correlations were initially computed and Fisher-z transformed. Subsequently, they were tested via a linear model that included state and trait anxiety scores, as well as differences in button presses. In the estimation of the posterior distributions, the effect of controllability was

captured by the model's intercept. Additional slope parameters evaluate the contributions of the covariates. Results were summarized in terms of $P+$ values, as defined previously.

## Data availability

The datasets generated during analyses during the currenty study are available in the LCE-UMD Github respository, https://github.com/LCE-UMD/controllabilty_paper.

## Code availability

Scripts to reproduce data analyses and generate plots can be found on our Github repository for this manuscript: https://github.com/LCE-UMD/controllabilty_paper.

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

## Acknowledgements

Research was supported by the National Institute of Mental Health (R01 MH071589 and R01 MH112517). We thank Gang Chen for discussions concerning statistical analysis, including Bayesian multilevel approaches.

## Author contributions

L.P., C.M., and S.P. designed the experiment; A.K. collected participant data with initial help from C.M.; C.L. performed and optimized statistical analysis while S.P. and C.M. performed more preliminary analyses; K.M. worked on data visualization; K.M. and A.K. generated figures; L.P., C.L., and K.M. interpreted results; and L.P. drafted the manuscript.

## Competing interests

The authors declare no competing interests.
