## [Peer Review File · Communications Biology]

Reviewers' comments:

Reviewer #1 (Remarks to the Author):

In the present study, Limbachia and colleagues utilized Bayesian multilevel modeling to investigate fMRI signal responses to controllable and uncontrollable stress in humans during their recently development moving circles task. Regions of interest from key nodes involved in threat processing were included in the analysis with supplemental voxelwise analyses. Strong evidence of a controllability effect in the bed nucleus of the stria terminalis (BNST) was observed with other effects observed in the insula, thalamus, and amygdala. The study is important and contains information valuable to our understanding of the role controllability has on several brain regions important for healthy emotional function. Further, the authors employ novel analytic techniques very relevant to the neuroimaging field. This is a strong paper that nonetheless has some weaknesses that should be addressed. I have several comments and suggestions for the authors, most of which deal with the clarity of the methodology for interpretation.

1. The introduction is missing some important background work on prior human imaging work that investigated fMRI signal responses to threat. For example, though the authors consider the ventromedial PFC as an important region for controllability, relevant research (e.g., Wood et al., 2015 which also showed controllability in more posterior hippocampus) is omitted. Similarly, although prior research on stressor controllability, it is not clear what the prior work found. I would recommend better introducing readers to this prior work to better frame the relevance of the current study to prior literature.

2. The methodology could use some clarification. The authors note that they performed their analyses using "unsmoothed functional data" composed of one representative timeseries for each ROI (e.g., line 534). However, the methods and supplement note data was originally processed in 3ddeconvolve with a standard GV-HRF. The output of this is a voxelwise measure of amplitude, and thus it's unclear where the timeseries comes from. Further, the supplement describes a spatial smoothing step (4mm FWHM) prior to deconvolution. It is difficult to follow the specific steps the authors took to get to the final analyses and results. Clarification of the specific processing and analytics steps is needed.

3. Related to the above, I find the use of the FIR to plot the data somewhat confusing as the authors are essentially showing data they did not test. Although I understand the authors emphasize that these data are for visualization purposes and not statistical analyses, these data are still used to make inferences about neural responses. Could the analysis not have been done with the FIR model (and % signal change indexed as the amplitude at X time) instead of assuming the HRF for the main analysis?

4. The authors utilized ICA-AROMA to filter subject-level noise from the fMRI data. I also noticed however that head motion information was included as a regressor in the 1st-level models. This seems like a good way of safely dealing with motion, however some users of ICA-AROMA have noted that introducing certain regressors after ICA-AROMA can effectively reintroduce such noise into the data (please see relevant links from <https://neurostars.org/t/ica-aroma-agg-vs-non-agg/3708> and <https://neurostars.org/t/testing-different-noise-models-output-from-fmriprep-unclear-results-with-ica-aroma/2352> and note this has also been observed with data filtering steps as well), particularly the translation parameters typically derived from volume registration.

5. From the methods, I gathered that the present study is an analysis of a dataset the authors have published on previously. However, key aspects of the methodology seem to be put into the supplement. Information about the participants included in the manuscript should be placed in the main text. Likewise, the overlap in participants between the present study and the prior work should be noted as this is relevant to interpretation of the data.

6. The BLM approach has significant benefits and the field should be enthusiastic about alternative approaches to NHST. Further, it is clear that the authors have spent much time thinking through the application of this approach to the data and this is a tremendous plus. The thought of not needing to perform multiple comparison correction is a significant advantage. However, one minor issue is the interpretation. More clarity on $P+$ closer to 0 and how the authors chose which effects to discuss versus not. Some of this is discussed in the supplementary methods where the authors note that $P+$ values closer to 1 are more indicative of an effect than values closer to 0, and further argue for a lack of cut offs to denote strong, moderate, and weak effects and instead to "quantify and qualify the evidence". The issue, however, is the authors then go on to utilize this language in a way that's difficult to gauge (e.g., lines 272 and 273 discussing the effect of controllability on BST and amygdala responses). How are the authors defining what a strong versus moderate effect is in the current study? Should $P+$ be indicative of the "effect size" for interpretation? The effect of state anxiety on BST responses by the authors' metrics appears to be impactful ($P+ = 0.891$) yet this is seen as a "limited effect." It is also somewhat confusing how the $P+$ values from BLM are below 0 in some of the figures. The techniques the authors are employing are still relatively novel and better clarity of this would be appreciated.

7. The present study challenges prior models of the effect of controllability and the neural processes that guide responses to controllable and uncontrollable stressors. The present study is not a connectivity study that tested potential directionality of inputs/outputs from the MR signal so it is difficult to compare these results as the authors note. I wonder, however, if the authors could speculate a bit more as to how they believe the different neural responses co-occur within the general emotion regulation network. If the authors do not think the current results support prior models, what do they think is the processing stream for their effects?

Minor points

1. I would suggest the contrast is "controllable" vs "uncontrollable" stress as it's not clear that every response was necessarily "controlled."
2. Please clarify for data whether the reports are mean and standard deviation or standard error of the mean.
3. The authors note that they mean-centered the button press regressor to deal with potential multicollinearity with the stressor regressor. Wouldn't subtracting a constant from one variable leave the correlation between it and another variable the same? Mean-centering generally does not lessen the correlation between lower-order terms.
4. Labels for figures, particularly plots of neural responses (are these % signal change and time?) would be appreciated.
5. The authors note a study by "Knight and colleagues" but cite Harnett et al., 2015. Is this the correct citation?

References

Wood, K. H., Wheelock, M. D., Shumen, J. R., Bowen, K. H., Ver Hoef, L. W., & Knight, D. C. (2015). Controllability modulates the neural response to predictable but not unpredictable threat in humans. *NeuroImage*, 119, 371-381.

Reviewer #2 (Remarks to the Author):

Overview: This study aims to characterize the neural circuits underlying threat responses that are controllable vs. uncontrollable. Participants were scanned using fMRI during a task designed by the authors ('moving-circles paradigm'). During the task, a paired electric shock and aversive sound was delivered when two moving circles touched. Participants in the control group could terminate this stressor by making a number of button presses, while participants in the (yoked) uncontrollable group could make button presses that did not control stressor cessation. The authors use Bayesian multilevel analyses and standard voxel-wise analysis, focusing their analyses

to a series of ROIs implicated in threat processing and regulation. When comparing controllable threat responses to uncontrollable ones, the authors report reduced differences in bilateral activation of the BST and left dorsal ACC, as well as weaker activation in portions of the amygdala and thalamus (and other threat-related regions). They also report increased activation of PCC, medial frontal gyrus and posterior insula in exploratory analysis. This question is relevant as controllability has emerged as a primary modulator of anxiety and threat processing. My primary concerns regard the lack of information regarding the task structure and how the delivery of the aversive outcome was determined, as well as how this study is novel relative to previous work the authors cite in the introduction:

(1) It is unclear what the purpose was of having the circles occasionally touch to trigger the aversive outcome if the primary question in this study was how stressor controllability modulates neural responses. Can the authors elaborate on why they introduced such variability/uncertainty in the stressor delivery? One purpose that seems immediately clear is tracking how proximity of threat shapes neural responses, but here there are no analyses or discussion related to this point. The authors should provide more explanation of this choice aside from aligning the task with the rodent version of the study.

(2) Given that the circles touch no more than 3 times per block, was the movement of the circles predictable in any way or were they random? What was the likelihood that the circles would touch after a given interval of time? Is it possible for participants to learn the probability or patterns of circles touching, and therefore receiving the aversive outcome? The authors should provide evidence or clarity regarding how learning is controlled for in the study.

(3) It appears that there could be some interval of time before participants in the controllable group terminated the shock. How was the shock delivered during this time—was it delivered continuously for a number of seconds (if so, on average how long?) or was it delivered in repeated pulses? I assume the aversive sound was delivered continuously with the shock delivery? Regarding the SCRs, the authors report trial-by-trial correlations between SCR and brain responses but were there any differences between controllable/uncontrollable groups in terms of skin conductance response to the aversive outcome itself?

(4) The introduction cites 4 studies (5,6,10,12) that assess behavior and/or brain imaging during active avoidance/escape/controllability paradigms, but they do not address how their study differs from these previous investigations, thus it is difficult to ascertain the novelty of the study. While the Bayesian analysis approach is unique, and the authors highlight a characterization of the BST/PAG as a novel contribution, they should describe in more detail (i.e., a paragraph in the discussion) how their task and findings differ than previous work.

(5) Why was the left insula analyzed separately from the other 24 ROIs (Fig 6)?

(6) The authors suggest a lack of evidence to support the Meier 2015 model [LC and BST/dorsal raphe/amygdala and PAG]. How did their task differ from that which motivated Meier's model? The varying threat proximity and uncertainty inherent in the task may have contributed to the pattern of threat responses featured here and should be discussed in more detail in the discussion.

Minor points:

(1) What do the authors mean by "positive-" and "negative-going responses". This was unclear to me and is used a few times throughout the manuscript.

(2) What instructions were the participants given for controllable vs. uncontrollable groups?

Please find the revised version of our manuscript COMMSBIO-20-1930 "Controllability over stressor decreases responses in key threat-related brain areas". We are very grateful for the reviewers for the constructive feedback. Among the changes outlined below, (1) we have more clearly motivated the contribution of the paper; (2) clarified our statistical approach and methodological points; (3) analyzed now both the left and right anterior insula in a Bayesian multilevel voxelwise manner.

It's cliché to say it, but the paper is clearly improved now, at least we hope! We outline our point-by-point response below. The text in red font in the manuscript has been revised to address the critiques.

Reviewer #1

- *1. The introduction is missing some important background work on prior human imaging work that investigated fMRI signal responses to threat. For example, though the authors consider the ventromedial PFC as an important region for controllability, relevant research (e.g., Wood et al., 2015 which also showed controllability in more posterior hippocampus) is omitted. Similarly, although prior research on stressor controllability, it is not clear what the prior work found. I would recommend better introducing readers to this prior work to better frame the relevance of the current study to prior literature.*

We have reworked the Introduction to better discuss the study's background, and have included the work by Wood et al. (2015). See lines **51-66**. We also moved the discussion of the ROIs to the Results (lines **137-154**) to help focus the Introduction.

- *2. The methodology could use some clarification. The authors note that they performed their analyses using "unsmoothed functional data" composed of one representative timeseries for each ROI (e.g., line 534). However, the methods and supplement note data was originally processed in 3ddeconvolve with a standard GV-HRF. The output of this is a voxelwise measure of amplitude, and thus it's unclear where the timeseries comes from. Further, the supplement describes a spatial smoothing step (4mm FWHM) prior to deconvolution. It is difficult to follow the specific steps the authors took to get to the final analyses and results. Clarification of the specific processing and analytics steps is needed.*

The use of 3ddeconvolve with a "-1d" input option allows parameter estimation of a single time series, such as the unsmoothed but averaged-across voxels time series for an ROI. We have clarified that smoothing was only employed with voxelwise analysis (lines **488-492**).

- *3. Related to the above, I find the use of the FIR to plot the data somewhat confusing as the authors are essentially showing data they did not test. Although I understand the authors emphasize that these data are for visualization purposes and not statistical analyses, these data are still used to make inferences about neural responses. Could the analysis not have been done*

with the FIR model (and % signal change indexed as the amplitude at X time) instead of assuming the HRF for the main analysis?

We sought to estimate the effect of controllability in the most statistically efficient manner, which was especially important given that our design was not based on simple discrete events. This consideration determined our approach to data analysis. The second consideration is that we believe that visualizing responses is very important in fMRI studies, which often focus on activation maps. Together, we believe our approach is reasonable, that is, showing estimating responses without assuming a canonical hemodynamic response. We explain our rationale **lines 169-175**.

- *4. The authors utilized ICA-AROMA to filter subject-level noise from the fMRI data. I also noticed however that head motion information was included as a regressor in the 1st-level models. This seems like a good way of safely dealing with motion, however some users of ICA-AROMA have noted that introducing certain regressors after ICA-AROMA can effectively reintroduce such noise into the data (please see relevant links from <https://neurostars.org/t/ica-aroma-agg-vs-non-agg/3708> and <https://neurostars.org/t/testing-different-noise-models-output-from-fmriprep-unclear-results-with-ica-aroma/2352> and note this has also been observed with data filtering steps as well), particularly the translation parameters typically derived from volume registration.*

We verified that ICA-AROMA did not reintroduce head-motion noise into our data. Because we had head-motion regressors in the model, we can compare the 12 motion-related regressor coefficients with data applying ICA-AROMA and data without applying this step. If ICA-AROMA works as intended, the coefficients should be smaller on average when using this step relative to when not using it. Across all voxels within the ROIs investigated, we found that the summed values of the regressor coefficients (in absolute value) was 13.41 +/- 9.23 for the ICA-AROMA method and 18.04 +/- 13.79 without it. Thus, we saw a reduction on average motion-related components when ICA-AROMA was employed. Admittedly, the difference was relatively small but without ICA-AROMA the coefficients can be large at times. Critically, our model contains 12 motion-related regressor coefficients which absorb the undesirable contributions from head motion that remain after applying ICA-AROMA.

- *5. From the methods, I gathered that the present study is an analysis of a dataset the authors have published on previously. However, key aspects of the methodology seem to be put into the supplement. Information about the participants included in the manuscript should be placed in the main text. Likewise, the overlap in participants between the present study and the prior work should be noted as this is relevant to interpretation of the data.*

This is the first publication involving this dataset, although the paradigm is related to the one employed in a completely non-overlapping set of participants studied by us (Meyer et al., 2019). We moved information about participants to the main text (**lines 351-359**).

- *6. The BLM approach has significant benefits and the field should be enthusiastic about alternative approaches to NHST. Further, it is clear that the authors have spent much time thinking through the application of this approach to the data and this is a tremendous plus. The thought of not needing to perform multiple comparison correction is a significant advantage. However, one minor issue is the interpretation. More clarity on P+ closer to 0 and how the authors chose which effects to discuss versus not. Some of this is discussed in the supplementary methods where the authors note that P+ values closer to 1 are more indicative of an effect than values closer to 0, and further argue for a lack of cut offs to denote strong, moderate, and weak effects and instead to “quantify and qualify the evidence”. The issue, however, is the authors then go on to utilize this language in a way that’s difficult to gauge (e.g., lines 272 and 273 discussing the effect of controllability on BST and amygdala responses). How are the authors defining what a strong versus moderate effect is in the current study? Should P+ be indicative of the “effect size” for interpretation? The effect of state anxiety on BST responses by the authors metrics appears to be impactful (P+ = 0.891) yet this is seen as a “limited effect.” It is also somewhat confusing how the P+ values from BLM are below 0 in some of the figures. The techniques the authors are employing are still relatively novel and better clarity of this would be appreciated.*

We have expanded on the on the interpretation of the statistical evidence. The quantitative part involves the posterior distributions and P+ values. The qualitative part is further discussed in the Methods (**lines 556-563**). In addition, we clarified the interpretation of P+ values (**lines 119-121**): these probability values can only be between 0 and 1. For a contrast of Uncontrollable vs. Controllable: P+ close to 1 indicates Uncontrollable > Controllable, and P+ close to 0 indicates Uncontrollable < Controllable. We also clarified the caption to Figures 3 and 5.

- *7. The present study challenges prior models of the effect of controllability and the neural processes that guide responses to controllable and uncontrollable stressors. The present study is not a connectivity study that tested potential directionality of inputs/outputs from the MR signal so it is difficult to compare these results as the authors note. I wonder, however, if the authors could speculate a bit more as to how they believe the different neural responses co-occur within the general emotion regulation network. If the authors do not think the current results support prior models, what do they think is the processing stream for their effects?*

Essentially, we believe the three sites we uncovered (medial frontal gyrus, posterior cingulate cortex, and posterior insula) play a similar role as attributed to the ventromedial PFC when control is possible. We have elaborated on this point in **lines 278-282**.

Minor points

- *1. I would suggest the contrast is “controllable” vs “uncontrollable” stress as it’s not clear that every response was necessarily “controlled.”*

We have made the changes throughout the manuscript.

- *2. Please clarify for data whether the reports are mean and standard deviation or standard error of the mean.*

Reports are mean and standard deviations as now indicated (**line 130**).

- *3. The authors note that they mean-centered the button press regressor to deal with potential multicollinearity with the stressor regressor. Wouldn't subtracting a constant from one variable leave the correlation between it and another variable the same? Mean-centering generally does not lessen the correlation between lower-order terms.*

Mean-centering is an effective way to reduce the correlation between explanatory variables and is extensively used in multiple regression analysis. In the context of fMRI analysis, see for example Figure 5.11 of the Poldrack/Mumford/Nichols *Handbook of Functional MRI Data Analysis*.

- *4. Labels for figures, particularly plots of neural responses (are these % signal change and time?) would be appreciated.*

We have revised the plots to include the scales.

- *5. The authors note a study by "Knight and colleagues" but cite Harnett et al., 2015. Is this the correct citation?*

We have corrected the citation to Wood et al. (2015).

Reviewer #2

- *(1) It is unclear what the purpose was of having the circles occasionally touch to trigger the aversive outcome if the primary question in this study was how stressor controllability modulates neural responses. Can the authors elaborate on why they introduced such variability/uncertainty in the stressor delivery? One purpose that seems immediately clear is tracking how proximity of threat shapes neural responses, but here there are no analyses or discussion related to this point. The authors should provide more explanation of this choice aside from aligning the task with the rodent version of the study.*

The paradigm was developed with multiple analysis goals in mind, including related to threat proximity (for a similar paradigm, see Meyer et al., 2019) and threat dynamics (see Venkatesh et al., 2019). We will investigate these aspects in future analyses of the dataset. Here our focus was on the stressor delivery component and the role of controllability, as we explain in **lines 369-372**.

- *(2) Given that the circles touch no more than 3 times per block, was the movement of the circles predictable in any way or were they random? What was the likelihood that the circles would touch after a given interval of time? Is it possible for participants to learn the probability or patterns of circles touching, and therefore receiving the aversive outcome? The authors should provide evidence or clarity regarding how learning is controlled for in the study.*

To minimize any learning of motion patterns, participants experienced on average 4 stressors and 7 “near-misses” per run, so they could not determine when the circle approach would culminate in stressor delivery. (We defined a near-miss as the circles approaching each other and then retreating when the edges of the two circles were 1.5 or less circle diameters away.) In addition, because all participants experienced the same motion paths, the two groups were equated in this respect. We explain this in **lines 383-388**.

- *(3) It appears that there could be some interval of time before participants in the controllable group terminated the shock. How was the shock delivered during this time—was it delivered continuously for a number of seconds (if so, on average how long?) or was it delivered in repeated pulses? I assume the aversive sound was delivered continuously with the shock delivery? Regarding the SCRs, the authors report trial-by-trial correlations between SCR and brain responses but were there any differences between controllable/uncontrollable groups in term of skin conductance response to the aversive outcome itself?*

Median stressor delivery duration was 1.37 secs. We clarify stimulation administration in **lines 392-397**. Regarding SCR responses, we did not obtain evidence for a difference between stressor-related responses as a function of group ($P = 0.503$), as now described in **lines 193-194**.

- *(4) The introduction cites 4 studies (5,6,10,12) that assess behavior and/or brain imaging during active avoidance/escape/controllability paradigms, but they do not address how their study differs from these previous investigations, thus it is difficult to ascertain the novelty of the study. While the Bayesian analysis approach is unique, and the authors highlight a characterization of the BST/PAG as a novel contribution, they should describe in more detail (i.e., a paragraph in the discussion) how their task and findings differ than previous work.*

We further clarify our contribution in the Introduction (**lines 56-69**) and discuss them in more detail in the Discussion so as to elaborate on the contributions of the present work.

- *(5) Why was the left insula analyzed separately from the other 24 ROIs (Fig 6)?*

Presently, the Bayesian multilevel approach is not computationally feasible at the voxelwise, whole-brain level. Nevertheless, it can be extended to a voxelwise approach within anatomical territories. Accordingly, we extended the approach to analyze voxels of the insula (we now include results from both hemispheres), a key involved in threat/pain-related processing. We clarify these points in **lines 204-207**.

- (6) *The authors suggest a lack of evidence to support the Meier 2015 model [LC and BST->dorsal raphe->amygdala and PAG]. How did their task differ from that which motivated Meier's model? The varying threat proximity and uncertainty inherent in the task may have contributed to the pattern of threat responses featured here and should be discussed in more detail in the discussion.*

We agree that comparison with the Meier model is not straightforward. Nevertheless, the model is important enough that we discuss our findings in the context of their model in a more general manner (**lines 298-309**).

Minor points:

- (1) *What do the authors mean by "positive-" and "negative-going responses". This was unclear to me and is used a few times throughout the manuscript.*

We simply meant increasing/decreasing relative to baseline, as explained now in **lines 169-175**.

- (2) *What instructions were the participants given for controllable vs. uncontrollable groups?*

Controllable participants were told that the duration of shock could be reduced if they pressed the button frequently and fast enough (**lines 414-415**). Uncontrollable participants were told that they were asked to press the button to test their motor responding (**lines 428-429**).

REVIEWERS' COMMENTS:

Reviewer #1 (Remarks to the Author):

The authors have addressed my major concerns with satisfactory edits. The revised manuscript would be a nice addition to the literature.

Reviewer #2 (Remarks to the Author):

The authors have adequately addressed my concerns.